# Fouling and Performance Investigation of Membrane Distillation at Elevated Recoveries for Seawater Desalination and Wastewater Reclamation

**DOI:** 10.3390/membranes12100951

**Published:** 2022-09-28

**Authors:** Abdulaziz Khan, Sudesh Yadav, Ibrar Ibrar, Raed A. Al Juboori, Sara Ali Razzak, Priyamjeet Deka, Senthilmurugan Subbiah, Shreyansh Shah

**Affiliations:** 1Centre for Green Technology, School of Civil and Environmental Engineering, University of Technology Sydney, 15 Broadway, Ultimo, NSW 2007, Australia; 2Mechanical Department at Taif Technical College, Technical and Vocational Training Corporation (TVTC), Riyadh 11564, Saudi Arabia; 3NYUAD Water Research Centre, New York University, Abu Dhabi Campus, Abu Dhabi P.O. Box 129188, United Arab Emirates; 4Medical Physics Department, Al-Mustaqbal University College, Babylon 51001, Iraq; 5Department of Chemical Engineering, Indian Institute of Technology Guwahati, Guwahati 781039, Assam, India; 6Lexcru Water Tech Pvt. Ltd., Ahmeadabad 382418, Gujarat, India

**Keywords:** membrane distillation, membrane wetting, membrane fouling, AGMD, elevated recovery

## Abstract

This study reports on the impact of elevated recovery (i.e., 80%, 85%, and 90%) on the fouling and performance of air gap membrane distillation (AGMD) with real seawater and landfill leachate wastewater samples using polytetrafluoroethylene (PTFE) polymer membranes. Increasing the feed temperature from 55 °C to 65 °C improved the water flux of seawater and wastewater and shortened the operating time by 42.8% for all recoveries. The average water flux in the 80%, 85%, and 90% recovery experiments at the 65 °C feed temperature was 32%, 37.32%, and 36.7% higher than the case of 55 °C for the same recoveries. The water flux decline was more severe at a higher temperature and recovery. The highest flux decline was observed with a 90% recovery at 65 °C feed temperature, followed by an 85% recovery at 65 °C. Close examination of the foulants layer revealed that seawater formed a cake fouling layer made predominantly of metal oxides. In contrast, the landfill leachate fouling was a combination of pore blocking and cake formation, consisting mainly of carbonous and nitrogenous compounds. Physical cleaning with deionized (DI) water at 55 °C and 65 °C and chemical cleaning with hydrogen peroxide (H_2_O_2_) were investigated for their efficiency in removing membrane foulants. Analytical results revealed that seawater fouling caused membrane pore blockage while wastewater fouling formed a porous layer on the membrane surface. The results showed that membrane cleaning with hydrogen peroxide restored >97% of the water flux. Interestingly, the fouling factor in seawater tests was 10%, while it was 16% for the wastewater tests.

## 1. Introduction

Water scarcity predictions in the coming decades have led to a surge in research on seawater desalination and wastewater reclamation [1]. Pressure-driven membrane filtration, such as reverse osmosis (RO) for seawater and wastewater treatment, is the most common method for freshwater supply in arid regions [2,3,4,5,6,7]. The RO technology can remove more than 99% of dissolved ions and achieve up to 50% and 75% recoveries from seawater desalination [2,3,8,9] and brackish water treatment [10], respectively. Recent advancement in RO has led to the development of osmosis-assisted reverse osmosis (OARO), which can achieve a recovery of 80% [11]. Despite the numerous advantages of RO, it is still an energy-intensive process that uses high-grade energy (electricity) and suffers membrane (bio)fouling that affects its performance over time. Concerning seawater treatment, a feed with a medium salinity requires an energy of 2.3 to 4.0 kWh/m^3^ [12]. The membrane fouling by organic and inorganic water constituents is a common issue in the RO process [13,14]. Furthermore, it is graver in wastewater reclamation due to higher concentrations of different organic residuals, requiring more frequent membrane cleaning and premature module replacement [13]. Intensive feed water pretreatment reduces membrane fouling at the expense of higher operating costs [15,16]. Nevertheless, increased treatment costs make pressure-driven membrane technologies less attractive for water reclamation.

An emerging process for freshwater production, membrane distillation (MD), is a potential alternative for RO technology with multiple merits, such as low fouling propensity and high recovery [17]. A recent comprehensive review of membrane desalination processes showed that MD is the only technology comparable to RO in terms of performance and rejection [18]. In the past decade, the application of MD for wastewater treatment and seawater desalination has gained tremendous interest [19]. Among different MD configurations, air gap MD (AGMD) is one of the MD configurations that has a great potential to be exploited on a full scale due to its lower heat losses and higher energy efficiency while maintaining high permeate water quality.

Most AGMD studies are conducted using synthetic sea and wastewater as feed, which makes the interpretation of research data and AGMD performance hard to compare to the real-life situation of the industry due to the complicated composition existing in natural water sources. While the AGMD process utilizing real sea and ground waters is also a common research topic [20], assessment of AGMD applicability for treating wastewater, including landfill leachate, is scarce. The leachate produced from the commonly practiced solid waste management processes, namely landfill, contains hazardous contaminants such as ammonia and heavy metals [21]. Hence, treating this wastewater is important for preventing soil and groundwater contamination [22] and, at the same time, recovering important resources such as water, nitrogen, and phosphorous [23].

The focus of AGMD investigations, in general, has mainly been directed towards improving process efficiency through optimization of operating parameters (theoretically and experimentally) [24], design improvement, or exploration of alternative heat sources [25] to make the technology more affordable [26]. Little attention has been paid to studying MD performance at high recoveries, especially for natural seawater and landfill leachate. Few studies investigated the AGMD performance at relatively high feed recovery. For example, Doug et al. [27] used the AGMD process for seawater desalination at 70% recovery. Furthermore, it has been pointed out that increasing water recovery increases the thermal efficiency of the MD process, but it leads to excessive fouling development [27]. Fouling, wetting, and mechanical robustness are three major challenges that face the MD process for seawater and wastewater treatment [3]. The severity of fouling and its nature at high recovery have not been investigated for seawater and landfill leachate.

Despite the large research output in AGMD technology, no study investigated the MD membrane fouling at elevated recoveries for wastewater reclamation. Most studies investigated the AGMD fouling at moderate recovery levels. High recovery is particularly important in desalination and wastewater reclamation to reduce brine and waste discharge. Additionally, the fouling mechanisms of the AGMD system vary with the feed type, and there are no data to compare seawater and wastewater fouling mechanisms at elevated recoveries. In our study, we systematically assessed AGMD performance at high recoveries (80%, 85%, and 90%) by utilizing two types of feed waters with very different organic and inorganic contents: seawater and landfill leachate. The AGMD experiments were conducted under similar operating conditions and careful analysis of the fouling layer using a range of analytical tools such as field emission scanning electron microscopy (FE-SEM), energy dispersive spectroscopy (EDS), Fourier transform infrared (FT-IR) spectroscopy, contact angle, and porosity analysis. Finally, H_2_O_2_ was proposed for efficient fouling removal and permeate flux restoration.

## 2. Material and Methods

### 2.1. Feedwaters and Chemicals

Seawater and landfill leachate were used in the AGMD experiments as feed water. Seawater was collected from Bondi Beach in Sydney (Australia). The biologically treated landfill leachate was obtained from the Whyte Gully Landfill in New South Wales, Australia, stored at 6 °C, and used without dilution. Table 1 shows the characteristics of seawater and leachate samples. The Whyte Gully Landfill provided concentrations of ammonia, total suspended solids (TSS), and total irons. The other parameters were measured using standard methods [28]. A 30% hydrogen peroxide (H_2_O_2_) mixture was obtained from Merck, Australia, and the solution was diluted to 3% with DI water and used for membrane chemical cleaning.

### 2.2. PTFE Membrane Specifications

The Membrane Solutions (Shanghai, China) provided the polytetrafluoroethyleneAub (PTFE) membrane throughout the experiments. The membrane has good thermal stability and mechanical durability specifications to resist heat [20]. The main characteristics of the membrane are shown in Table 2.

### 2.3. AGMD Module Setup and Experimental Methodology

The AGMD module was made from Plexiglas to allow for visual observations. The dimensions of the AGMD module are as follows: 180 mm × 44 mm × 130 mm (length × depth × height). A schematic illustration of the AGMD module is shown in Figure 1. The picture of the AGMD setup used is also given in the Appendix A (Figure A1). Each part of the feed and coolant side has an inlet and an outlet port. The total area of the spacer frame from the coolant side was 0.0084 m^2^, and the membrane with an effective surface area of 0.0045 m^2^ was installed on one side of the spacer frame. The feed temperatures were set to 55 °C or 65 °C, and the coolant temperature was maintained at 18 °C. The flow rates of the feed and coolant solutions were 2.4 L/min and 1.6 L/min, respectively, and the flow rate was measured using panel mount flow meters (Blue White, Sterlitech, Auburn, AL, USA). The feed solution was placed on a hot plate (Guardian 500 Hotplate, Instrument Choice, Sydney, Australia) with a thermostat to control the temperature of the feed solution.

Each AGMD experiment was started by measuring the initial water flux of pristine membrane with DI water at 55 °C and 65 °C for 2 h. Then, seawater or leachate wastewater at 55 °C or 65 °C was supplied to the AGMD module, and experiments were continued until 80%, 85%, or 90% recovery was reached.

The permeate flux (J) was calculated following the expression:(1)$$\;J=\frac{V}{A\times \Delta t}$$
where *J* is the permeate flux, L/m^2^h; *V* is the permeate volume, L; *A* is the active membrane area, m^2^; *t* is time, hours.

After seawater/landfill leachate experiments, the membrane was cleaned with DI water at 55 °C or 65 °C for 1 h, and the water flux of the DI water was measured.

The fouling factor (*FF%*) of the membrane after cleaning was calculated as follows:(2)$$FF=\frac{J_{i}-J_{a}}{J_{i}}\times 100$$

In Equation (2), *J_i_* is the initial DI permeate flux before seawater/leachate wastewater treatment, and *J_a_* is the DI water flux after fouling and before chemical cleaning. The fouled membrane was cleaned with a 3% H_2_O_2_ solution for 30 min at 22 °C.

After H_2_O_2_ cleaning, the PTFE membrane was tested for salt rejection using a 35 g/L NaCl solution. The rejection (*R*%) was calculated using Equation (3):(3)$$R=\left(1-\frac{C_{p}}{C_{f}}\right)\times 100$$
where *(R%)* is the rejection, *C_p_* is the salt concentration in the permeate (mg/L), and *C_f_* is the salt concentration in feed water (mg/L). The salt concentrations were measured using TDS measurements.

The recovery (*Rr%*) is calculated as the ratio of permeate flow to the feed flow according to the following expression:(4)$$Rr=\frac{Qp}{Qf\;}100\%$$
where *Q_p_* and *Q_f_* are the permeate and feed solutions flow rates (L/h), respectively.

### 2.4. Membrane Characterizations

#### 2.4.1. FE-SEM and EDX Analysis

FE-SEM and EDS characterized the pristine and fouled membranes to obtain qualitative and quantitative information about membrane foulants. The FE-SEM was conducted with a Zeiss Supra 55VP SEM (Carl Zeiss AG, Oberkochen, Germany) with a Schottky source, and the accelerating voltage was set to 3 kV. An Oxford detector was used to conduct the EDS analysis of the fouled, cleaned, and pristine membranes. Membrane samples were dried at ambient temperature for 48 h in a clean room and then double coated with a gold layer.

#### 2.4.2. Pore Size and Contact Angle Analysis

The membrane pores and contact angles of virgin and fouled membranes were measured twice to monitor the changes in their average pore size and hydrophobicity after treatment with seawater or landfill leachate wastewater. The pore sizes of the pristine and fouled PTFE membranes were measured using the Techporo-AL-500 from Tech Inc Technologies(Chennai, India); more information about the method is available in Appendix A, Figure A1. The contact angle was measured by the sessile drop method utilizing CAM101 Contact Angle Analyzer (KSV instruments, Helsinki, Finland). The contact angle measurement was performed on different areas of the sample membrane, and the averaged values are reported in this study.

#### 2.4.3. FT-IR Analysis

FT-IR was conducted for functional group analysis of the pristine and fouled membranes using a Thermo Scientific Nicolet 6700 FT-IR (ThermoScientific, Sydney, Australia) spectrometer. Each scan was performed at least two times.

## 3. Results and Discussion

### 3.1. Coupled Effects of Temperature and Recovery on Seawater Treatment

The average DI water flux measured at 55 °C and 65 °C was 18.69 ± 1.2 L/m^2^h and 30.47 ± 1.5 L/m^2^h, respectively. The feed temperature in the first set of experiments was set to 55 °C, and experiments were conducted until 80%, 85%, or 90% recovery was reached. The data points in Figure 2A show an increase in the accumulated permeate volume over the processing time. The highest accumulated permeate volume was 908.1 ± 5 mL at a 90% recovery, 880 ± 5 mL at 85% recovery, and 811 ± 5 mL at an 80% recovery. The average permeate flow rate was 81.13 ± 2 mL/h for the 80% recovery, and 76.17 ± 3 mL/h and 75.67 ± 2 mL/h for the 85% and 90% recoveries, respectively. This corresponded to average water fluxes of 18.03 L/m^2^h, 16.93 L/m^2^h, and 16.82 L/m^2^h at 80%, 85%, and 90% recoveries, respectively (Figure 2B). Results showed that the average water flux decreased with the recovery from 80% to 85%, followed by a slight increase in the average water flux when the recovery increased to 90%. The decrease in the flux is expected due to the increase in membrane fouling over time. Higher recovery leads to a higher feed solution concentration and a lower water flux in the long run. The results agree with previous studies that showed a decline in the water flux flow with the recovery increase due to concentration and thermal polarizations [29]. The 90% recovery took a shorter time; therefore, the average flux is higher.

The accumulative permeate volume and water flux patterns for experiments conducted with seawater at 65 °C are presented in Figure 2C,D. The increase in the accumulated permeate volume was sharper at 65 °C compared to 55 °C. The respective accumulative permeate volumes for 90%, 85%, and 80% recoveries were 953 ± 3 mL, 873 ± 3 mL, and 835 ± 3 mL. The average water flux in the AGMD unit was 26.52 ± 1 L/m^2^h, 25.81 ± 1 L/m^2^h, and 26.48 ± 1 L/m^2^h for 80%, 85% and 90%, respectively. The water flux for all recoveries drops at 8 h, indicating serious fouling development after this period of operation.

As expected, the average water flux was higher in the AGMD operated at 65 °C than at 55 °C feed temperature. The average water flux in the 80%, 85%, and 90% experiments at 65 °C feed temperature was 32%, 37.32%, and 36.7% higher than in the case of 55 °C. The results agree with previous studies, which demonstrated that higher feed temperatures lead to higher vapor pressure in the MD system and consequently increases the water flux [30]. However, the recovery increase to 90% did not seem to impact water flux, which is a positive sign. At 65 °C, the water flux decline was more severe than at 55 °C, as shown in Figure 2D. The results also agree with previous studies, where flux decline was more severe at 70 °C than 50 °C [31]. The water flux decline for all experiments followed a similar trend.

### 3.2. Membrane Fouling and Cleaning with Seawater Treatment

Membrane fouling is driven by a combination of foulant materials in seawater that can create a fouling layer on the membrane surface and/or block membrane pores, as discussed in Section 3.6. As shown in Table 1, seawater is rich in inorganic foulants, including a high concentration of divalent calcium and magnesium ions that is expected to exacerbate membrane fouling when reacting with the seawater organic matter. However, inorganic scaling would be common fouling in MD processes at high recoveries as the solubility limits of scaling salts are likely to be exceeded. Compared to the pristine membrane, the elemental composition of calcium and magnesium ions on the fouled membrane surface increased 14 and 3.6 times (Table 3), suggesting the precipitation of calcium and magnesium oxides and salts. Figure 2B,D show that the water flux dropped over time at both feed temperatures but was more severe at 65 °C feed temperature. The decline in water flux at 55 °C is insignificant. Figure 2B shows a relatively stable water flux up to 12 h of operation at 55 °C feed temperature. In contrast, for the 65 °C feed temperature, a water flux decline occurred after 7 h (Figure 2D). Membrane scaling increases with feed temperature increase from 55 °C to 65 °C, causing a sharp drop in the water flux [32].

Figure 3A,B show the permeate TDS for the two tested feed temperatures. The permeate TDS seems to be inversely correlated to the recovery. For 55 °C experimental tests, the highest permeate TDS was observed for the lowest recovery, whereas the lowest TDS was observed for the highest recovery of 90%. The fouling factor was almost the same at all recovery percentages. For the 65 °C experiments, the highest permeate TDS was observed for the 85% recovery due to membrane fouling in this experiment, which exhibited the highest fouling factor (Figure 4). The permeate TDS at 65 °C for 90% recovery was almost at the same level as that at 55 °C. This could be attributed to the higher water flux, resulting in diluting the permeate flow. Therefore, the 65 °C feed temperature is more desirable for seawater treatment to increase the water flux within a short time by 4–5 h.

The DI water at 55 °C or 65 °C (depending on the feed temperature of the experiment) and H_2_O_2_ solution were investigated for membrane cleaning at the end of each AGMD experiment. For the DI water cleaning, the PTFE membrane was flushed with DI water for 60 min, and water flux was measured to compare with the initial water flux. The H_2_O_2_ solution cleaning was for 30 min, and water flux and salt rejection were measured for comparison purposes. To evaluate the impact of H_2_O_2_ cleaning on the membrane, the membrane rejection of NaCl was measured after the H_2_O_2_ cleaning and compared with the initial rejection. Heated DI water would release fouling materials loosely attached to the membrane surface, including organic and inorganic matters [33], and formed at the early stage of the filtration process. Over time, the fouling layer becomes denser (Table 3) and more stubborn to remove due to the metal oxides and salt accumulation on the membrane surface, rendering the membrane to become less hydrophobic (Table 4). The drop in the water flux after DI water cleaning at 55 °C or 65 °C is attributed to the precipitation of metal oxides and salts (Table 4) that cannot be removed by simple hot DI water cleaning. Hot DI water has been proven effective for removing large amounts of foulants; however, it may lead to irreversible fouling [34]. The fouling factor (Figure 4) shows that the fouling percentage on the membrane surface at 55 °C and 65 °C and for all recoveries was 4% to 9%, respectively, which agrees with previous experimental work findings [35]. As reported in previous studies [2], the difference in membrane fouling between 55 °C and 65 °C decreased as the experimental time decreased. The permeate TDS was higher in the experiments performed at 65 °C than at 55 °C, indicating more severe pore wetting. Iron and magnesium hydroxide, for example, precipitate at higher feed temperatures and could be responsible for pore wetting and increased permeate TDS. Generally, the results demonstrate that flushing with hot DI water is not enough, while H_2_O_2_ was effective for membrane cleaning, and the membrane rejection was almost restored to the pristine membrane.

### 3.3. Coupled Effects of Temperature and Recovery on Wastewater Treatment

Accumulated permeate volume and water flux results for landfill leachate are presented in Figure 5. At 55 °C feed temperature, the average permeate volume and water flux for 80% recovery were 77.1 ± 3 mL and 17.13 ± 3 L/m^2^h, respectively. In contrast, for 85% recovery, they were 71.25 ± 2 mL and 15.83 ± 2 L/m^2^h, and for 90% recovery were 90.89 ± 2 mL and 20.2 ± 3 mL/h. Figure 5A,B show that the accumulated permeate volume increased gradually while the water flux decreased over time. The average permeate volume and water flux at feed temperature of 65 °C and 80% recovery was 131.83 ± 3 mL and 29.3 ± 3 L/m^2^h, while the average permeate volume and water flux for 85% was 127.21 ± 2 mL and 28.27 ± 2 L/m^2^h, and for 90% was 131.94 ± 2 mL and 29.32 ± 3 L/m^2^h. The water flux dropped over time, and the accumulated permeate volume rose gradually (Figure 5C,D). Compared to 55 °C tests, the 65 °C tests achieved 45.5%, 45.3, and 32.7% greater water flux for 80%, 85% and 90% recoveries, respectively. Additionally, the filtration cycle was shorter at 65 °C by 4–5 h than at 55 °C for the same recoveries. The increase in the temperature of the wastewater on the feed side by 10 °C accelerated the evaporation on the feeding solution side of the membrane, leading to more vapor transport through the membrane to the air gap, and hence higher water flux was achieved.

### 3.4. Membrane Fouling and Cleaning with Leachate Treatment

Similar to the seawater experiments, membrane cleaning with hot DI water for 60 min and then with H_2_O_2_ solution for 30 min was applied at the end of each experiment. The TDS concentration in the permeate increased over time, indicating a reduction of the PTFE rejection caused by membrane wetting due to fouling (Figure 6A,B) [27]. Organic fouling is dominant on the membrane after the leachate wastewater experiment, evidenced by the high elemental carbon on the fouled membrane surface (Table 3). The results in Table 4 revealed a sharp drop in contact angle of the PTFE membrane treating wastewater from 129° to 93° after 30 h, indicating fouling accumulation.

The rejection of the membrane after H_2_O_2_ cleaning was very close to that of the pristine membrane (~98%), indicating the high efficiency of the cleaning process. The membrane fouling factors were between 12% and 16% for 55 °C and 65 °C feed temperatures (Figure 7), while they were between 4% and 9% in the seawater experiment. The higher turbidity and organic matter concentration of wastewater feed solution (Table 1) resulted in severe membrane fouling (as discussed in Section 3.5). Additionally, the thermal breakdown of some organic matter in the feed solution to smaller molecules might have caused pore plugging. Earlier studies suggested that the degradation of organic matter led to membrane fouling and water flux decline [17]. The yellowish deposition in Figure 8D could be caused by organic and humic acid fouling [17]. This finding is supported by the high elemental C on the membrane surface used with landfill leachate, as shown in Table 3.

### 3.5. Membrane Characterization Tests

#### 3.5.1. FE-SEM, EDX, and FT-IR Analysis

Pristine and fouled membranes were examined with SEM and EDX techniques to visualize and analyze the fouling layer on the membrane surface (Figure 8 and Table 3). The SEM of the pristine membrane shows long fiber-like structures (Figure 8A). These structures are almost invisible in the fouled membranes, indicating the severity of the fouling on the membrane surface. Compared to the pristine membrane, the SEM analysis of the fouled seawater membrane shows a non-uniform cake layer of foulants and a small crystalline salt structure on the surface (red circle in Figure 8B), indicated by the small nodules-like structures in the middle. This is also evident from the EDX analysis, where the Na elemental composition shows the highest percentage for seawater-fouled membranes, followed by Mg (5.82%) and Ca (3.6%) (Table 3). Generally, the concentration of Na in the seawater is the highest among all the ions. The divalent ions in the seawater can also act as a bridge for organics in the seawater, promoting severe fouling on the membrane [30]. The presence of Cl^−^ is also evident from the EDX analysis, which may be due to the ions trapped in the pores of the PTFE membrane. This may indicate NaCl is trapped in the membrane’s pores since Na and Cl^−^ ions follow the same pattern (EDX, Table 3). The high presence of these inorganic elements can also promote membrane wettability and decrease membrane rejection compared to the pristine membrane. Other inorganic foulants, such as S, are also on the membrane surface. S can promote iron sulfide or magnesium sulfate fouling on the PTFE membrane. The EDX analysis shows the presence of metal oxides marked by increased O and metal concentrations. However, an increase in O can also be due to the carbonate or sulfate ions that precipitate as scaling when they form insoluble salts with Mg and Ca ions [36]. FT-IR analysis was used to examine the foulants layer structure, as shown in Figure 9. The sharp peak at 2916 cm^−1^ can be attributed to the saturated fatty acid chains due to the presence of lipids in ocean water [37]. The reduction in the characteristic peaks of the pristine membrane indicates the formation of a fouling layer [38]. The small peaks observed in the range of 1500–1600 cm^−1^ in a spectrum of the membrane fouled with landfill leachate suggest the presence of amine related to protein structure [39,40]. This agrees with the EDS analysis, which shows a high N percentage in the fouled membranes with landfill leachate.

There is also an increase in the intensity of peaks of 1458 (cm^−1^) and 1373 (cm^−1^) that is attributed to the C-H bending for the foulants in the seawater [41]. The SEM image of the PTFE membrane fouled by landfill leachate wastewater for 15 h shows a non-uniform but thick fouling layer compared to the seawater-fouled membrane (Figure 8C). The fouling is more intense on the PTFE membrane used for a 30 h test with landfill leachate (Figure 8D).

In both SEM images for the landfill leachate wastewater, the presence of inorganic salts is less than in the seawater-fouled membranes, as indicated by the elemental composition in the EDS analysis. However, 30 h fouled membrane shows more inorganic ions compared to a membrane fouled for 15 h. The presence of Fe (2.06%) is the highest for the membranes fouled by landfill leachate for 30 h, while the presence of N is predominant in 30 h fouled membranes (50.39%), followed by 15 h fouled membranes with landfill leachate. The FT-IR spectra of all the fouled membranes (Figure 9) show a decrease in the intensity of the membrane characteristic peaks compared to the pristine membrane, which is attributed to the coverage of the membrane surface by different foulants. The highest decrease in the intensities is observed for wastewater membrane operated at 90% recovery. The spectra of the membrane at 80% and 85% are almost similar; however, more fouling intensity is observed compared to seawater-fouled membranes. Another small peak is evident at 3750 cm ^−1^ for the wastewater-fouled membrane. These bands are attributed to the NH peaks associated with protein foulants [42].

#### 3.5.2. Pore Size and Contact Angle Analysis

The pristine PTFE membrane’s pore size and contact angles were compared to the fouled membranes, as presented in Table 4. The pristine membrane pore sizes were larger than those of the fouled membranes. The smallest pore diameters were 0.213 ± 0.010 µm and 0.201 ± 0.008 µm for pristine and seawater-fouled membranes, respectively, indicating an insignificant change to the smallest pore diameters exerted by membrane fouling. Likewise, the largest pore diameter decreased from 0.296 ± 0.009 µm to 0.294 µm after seawater treatment. Similar results for PTFE membranes fouled by seawater were also observed in the previous study, with membranes exposed to seawater for four weeks [43]. A significant change was observed in the pore diameter of the fouled membrane after 30 h of landfill leachate wastewater treatment for the landfill leachate wastewater. Thus, the smallest pore diameter decreased from 0.213 ± 0.010 µm to 0.166 ± 0.006 µm, significantly affecting the contact angle (Table 4). The largest pore diameter experienced a 58% reduction (from 0.248 ± 0.008 µm to 0.170 ± 0.008 µm) after 30 h, indicating severe pore size narrowing due to membrane fouling after landfill leachate treatment.

The contact angle of the pristine membrane was compared with the fouled membrane to assess the membrane’s hydrophobicity changes after fouling. A significant change in contact angle was observed for membranes fouled by landfill leachate wastewater after 30 h of the MD process. The water contact angle of the PFTE membrane decreased by 28% after landfill treatment. Although H_2_O_2_ cleaning is an effective method for organic and inorganic foulant removal [44], it was insufficient to restore the water flux at high recovery. After 30 h of AGMD process with leachate feed, membrane fouling becomes denser, and fouling layers are probably stacked over each other at high recoveries. For the seawater-fouled membrane, there was a slight change in the hydrophobicity of the membrane, as evident by the contact angle of 115 ± 30. In seawater experiments, water flux was almost completely restored after the membrane cleaning [45]. Additionally, the slight change in contact angle of seawater experiments indicates the effectiveness of the H_2_O_2_ method in cleaning the PTFE membrane. Overall, the contact angle decrease was still within the desired value of the MD process operation, which was affected by permeate conductivity.

### 3.6. Comparison of Seawater and Wastewater Fouling

A visible inspection of the seawater-fouled membrane shows insignificant membrane fouling compared to the pristine PTFE membrane (Figure 10A,B). The fouling factors on the seawater at 55 °C for 80% to 90% ranged from 4% to 9% (Figure 4). Additionally, the water fluxes were between 16 L/m^2^h to 18 L/m^2^h for all recovery rates. The water fluxes increased almost twice when the feed temperature was increased from 55 °C to 65 °C, and the fouling factor remained quite stable, as shown in Figure 10C. EDS results for the landfill leachate experiments indicate a potential combined organic and inorganic fouling, evidenced by the detection of C, Mg, and Ca on the membrane surface (Table 3). A common mechanism of organic–inorganic fouling is that organic substances are bound to the hydrophobic membrane surface while divalent ions work as bridging elements [46]. Cleaning with hot DI water was ineffective, as visible foulants were still on the membrane surface (Figure 10B). Chemical cleaning by H_2_O_2_ for 30 min on the fouled membrane was more effective than hot DI water; however, the membrane still showed visible fouling signs. There is no significant difference in the membrane rejection due to the recovery increase from 80% to 90%.

The fouling factor for seawater at 55 °C to 65 °C is lower than that of wastewater. Generally, the fouling factor was lower in the seawater tests than in the landfill tests due to the severity of PTFE membrane fouling in the landfill wastewater tests. At the highest recovery levels, the fouling factor of the wastewater is almost twice that of the seawater at the same temperature. As mentioned above, the high fouling factor for the leachate wastewater can be attributed to the combined nature of the organic and inorganic foulants in the wastewater feed stream. Divalent cations in the wastewater feed stream exacerbated the organic matter fouling on the PTFE membrane [47,48]. At a high recovery, this fouling is predominant due to the high-water permeation and the concentration of the foulants on the membrane surface. For 65 °C feed temperature tests, the wastewater fouling factor is also significantly higher, about three times, than the seawater fouling factor. Similar results of severe fouling at higher feed temperatures have been reported for other types of wastewater and are linked to forming a heterogenous fouling layer [36]. The porous fouling layer for the leachate wastewater membrane is also evident from the SEM analysis in Figure 8D. Overall, membrane fouling for leachate wastewater treatment is more tenacious than that for seawater due to the combined effect of organic and inorganic fouling (Table 3). For all recovery levels and 55 °C, the fouling factor is 36% to 187% higher in the leachate wastewater tests than in the seawater tests. The difference in the fouling factor between seawater and leachate wastewater was 216% to 388% at 65 °C, depending on the recovery. For leachate wastewater and 65 °C feed temperature, there is a trivial difference in the rejection between the different recovery levels. The rejection for wastewater at 55 °C and 65 °C was between 98% and 99%, and fouling factors were between 12% and 16% for all recovery levels.

## 4. Conclusions

PTFE MD membranes were tested with natural seawater and wastewater samples to assess the impact of high recovery levels on the process performance. The impact of temperature on the performance was also investigated. The AGMD process was tested to achieve high recoveries of 80%, 85%, and 90% for 55 °C and 65 °C feed temperatures. DI water and H_2_O_2_ were used as cleaning chemicals to remove fouling from the PTFE MD surface. The seawater results showed that water fluxes at 55 °C for 80–90% were 16–18 L/m^2^h, with fouling factors between 4% and 9% for all recoveries. In comparison, the flux at 65 °C increased by almost double to reach 26 to 28 L/m^2^h with a slightly less level of the fouling factors for all recoveries. The rejection was 96% to 98% for all recoveries. The wastewater results showed that the fluxes at 65 °C were 28 to 30 L/m^2^h for all recoveries with a fouling factor between 12% and 16%. A significant reduction in water fluxes (42.9%, 45.2%, and 32.3% for 80%, 85%, and 90% recoveries, respectively) with the same level of fouling factors was observed when the feed temperature decreased by 10 °C. The rejection was 98% to 99% for all recoveries at 55 °C. Increasing the feed temperature from 55 °C to 65 °C improved water flux during seawater and leachate tests and shortened the operating time by 42.8%. For all recoveries, the fouling factor was less than 10% when seawater was used as a feed. On the contrary, the fouling factor in AGMD experiments with the landfill leachate almost doubled. The 3% H_2_O_2_ could not remove landfill leachate stains from the membrane surface, suggesting that a stronger concentration or other chemicals may be required to remove this fouling type.

Future work should investigate the impact of membrane material on the performance of the AGMD for the treatment of seawater and wastewater at elevated recovery to identify the suitability of membrane material for such treatment settings. Additionally, long-term and large-scale tests of AGMD with natural seawater and wastewater samples are important for a realistic evaluation of the technology as a competitive process in the water treatment industry.

## Figures and Tables

**Figure 1 membranes-12-00951-f001:**
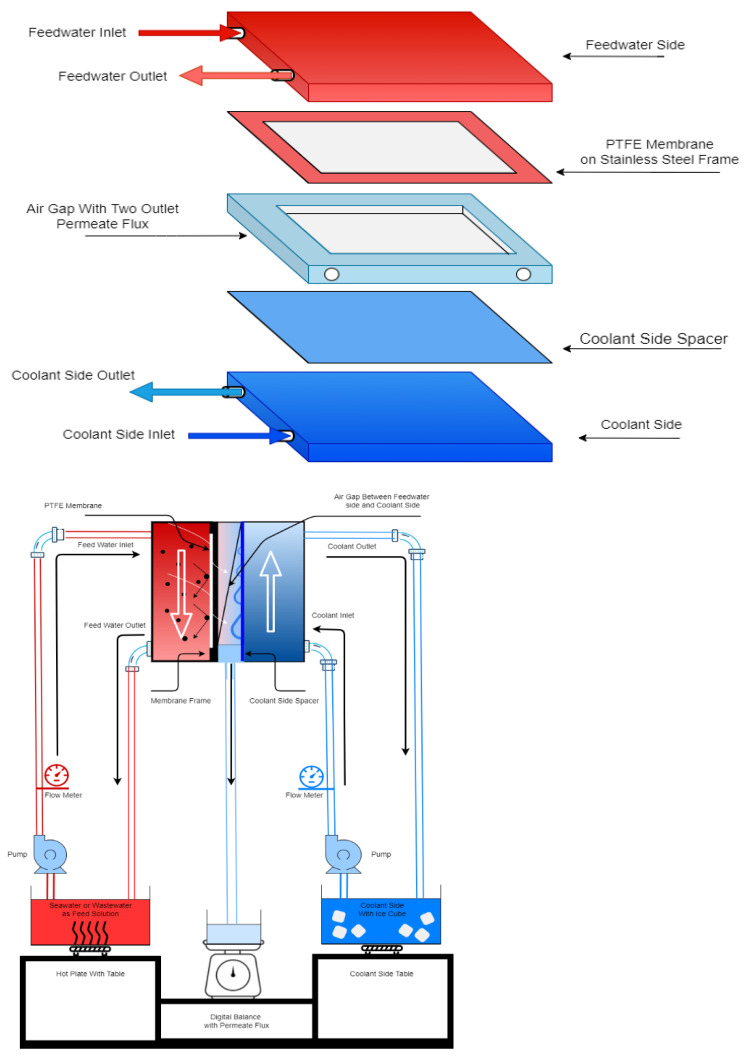
Schematic diagram of the AGMD unit.

**Figure 2 membranes-12-00951-f002:**
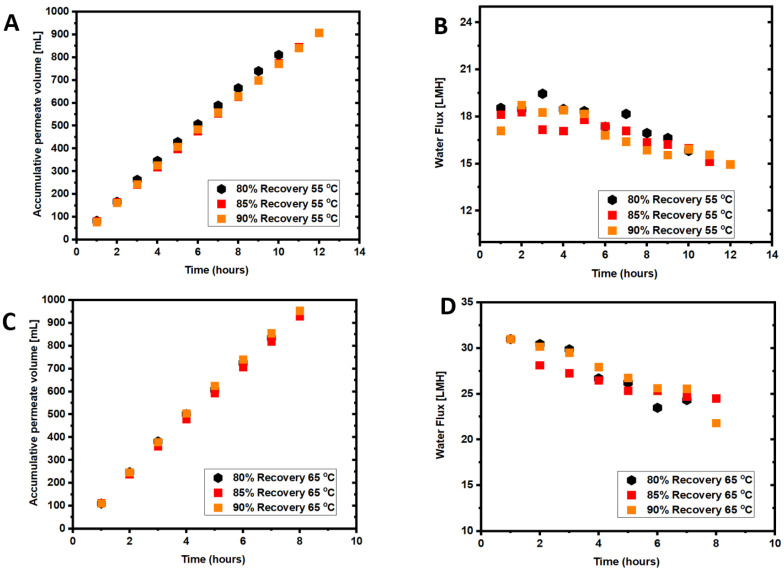
(**A**) Accumulative permeate volume at 55 °C, (**B**) water flux in the AGMD at 55 °C, (**C**) accumulative permeate volume at 65 °C, and (**D**) water flux at 65 °C. Feed solution: seawater (TDS = 32.8 g/L).

**Figure 3 membranes-12-00951-f003:**
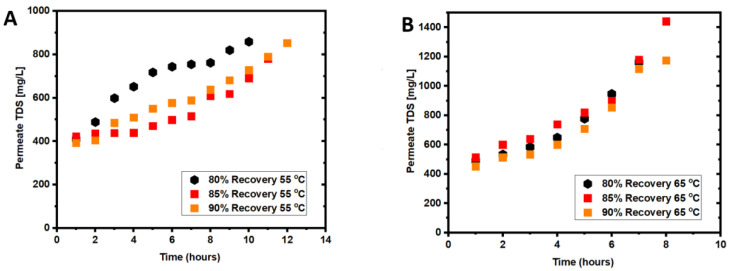
Permeate TDS for seawater treatment: (**A**) permeate TDS concentration for seawater treatment at 55 °C at 80%, 85%, and 90% recoveries; (**B**) permeate TDS concentration for seawater treatment at 65 °C at 80%, 85%, and 90% recoveries.

**Figure 4 membranes-12-00951-f004:**
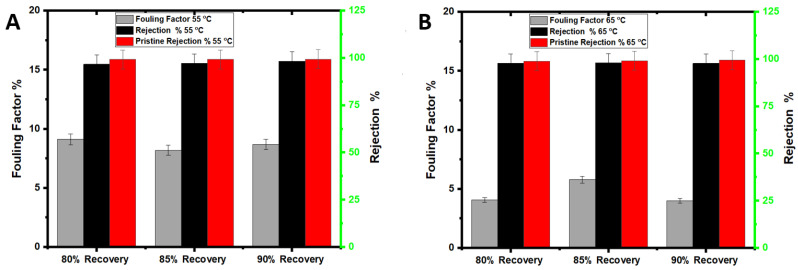
Fouling factor and rejection (**A**) at 55 °C and (**B**) at 65 °C.

**Figure 5 membranes-12-00951-f005:**
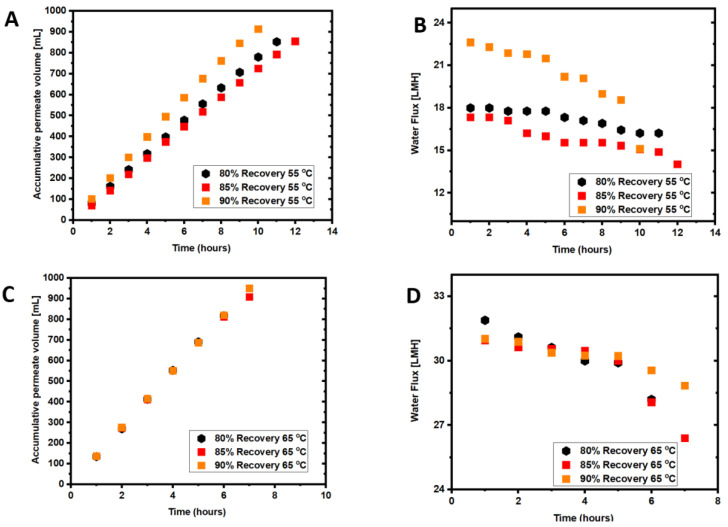
Permeate volume and water flux in AMGD: (**A**) 55 °C accumulative permeate volume; (**B**) 55 °C water flux in the AGMD; (**C**) 65 °C accumulative permeate volume, and (**D**) 65 °C water flux.

**Figure 6 membranes-12-00951-f006:**
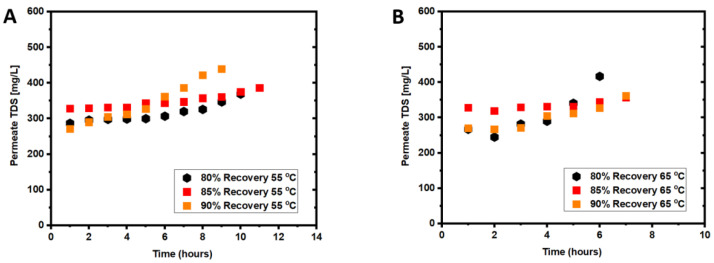
(**A**) Permeate TDS concentrations at 55 °C for 80%, 85%, and 90% recoveries; (**B**) permeate TDS concentrations at 65 °C for 80%, 85%, and 90% recoveries.

**Figure 7 membranes-12-00951-f007:**
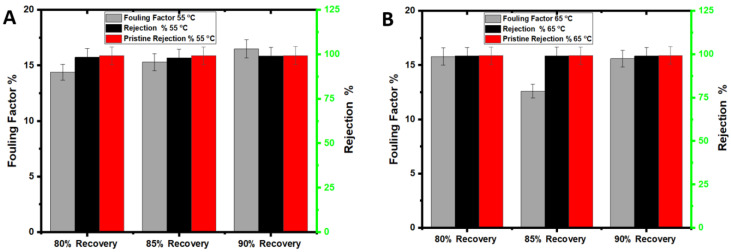
Fouling factor and rejection for wastewater treatment (**A**) at 55 °C and (**B**) at 65 °C.

**Figure 8 membranes-12-00951-f008:**
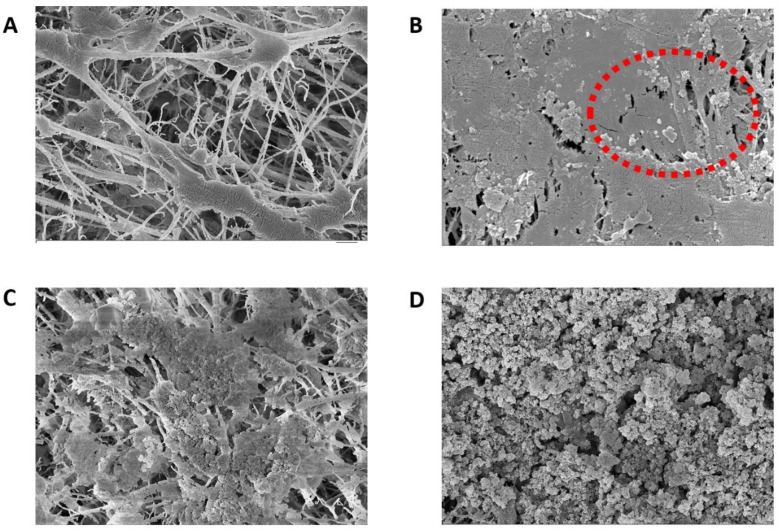
(**A**) SEM of pristine PTFE at 1 µm, (**B**) SEM of seawater-fouled PTFE membrane at µm, the red circle indicates inorganic scaling, (**C**) SEM of landfill leachate wastewater-fouled PTFE membrane for 15 h at 1 µm, and (**D**) SEM of membrane surface fouled with landfill leachate wastewater after 30 h of AGMD at 1 µm.

**Figure 9 membranes-12-00951-f009:**
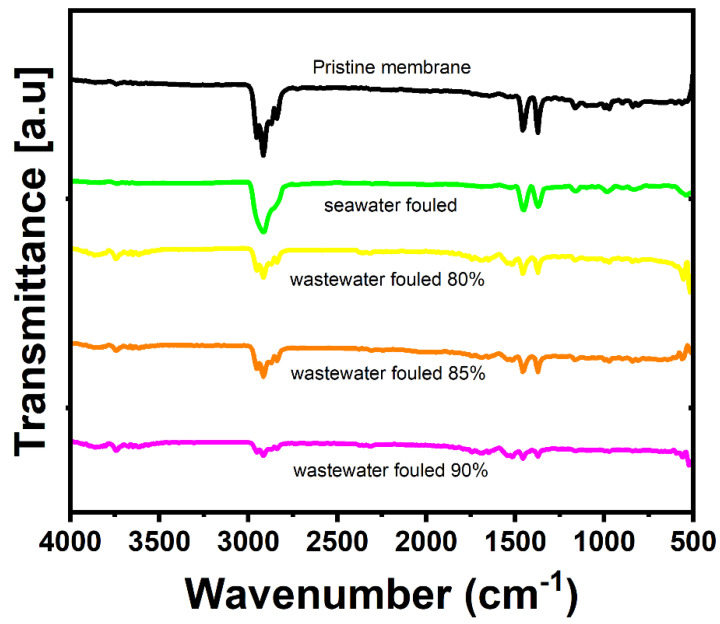
FT-IR bands of pristine and fouled membranes by seawater and wastewater at different recoveries.

**Figure 10 membranes-12-00951-f010:**
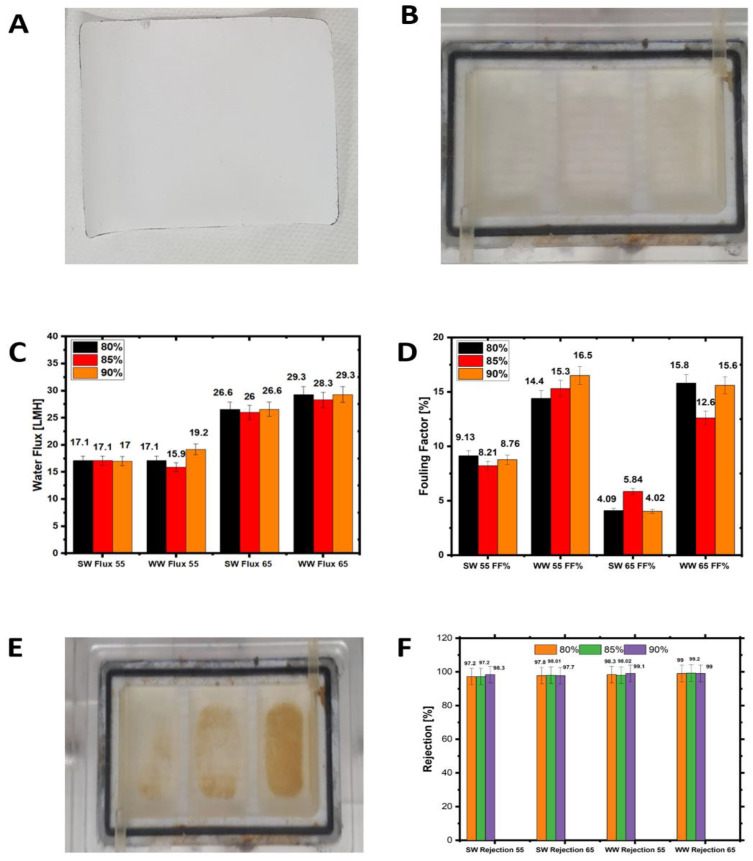
(**A**) Pristine membrane picture, (**B**) PTFE membrane after seawater treatment picture, (**C**) seawater average water flux with the fouling factors at 55 °C and 65 °C for all recoveries, (**D**) wastewater and seawater fouling factors at 55 °C and 65 °C for all recoveries, (**E**) picture of fouled wastewater membrane, and (**F**) comparison of seawater and wastewater rejection at 55 °C and 65 °C for all recovery rates.

**Table 1 membranes-12-00951-t001:** Characteristics of the seawater and landfill leachate.

Seawater			Landfill Leachate		
Parameter	Concentration (mg/L)	Measuring Instrument	Parameter	Concentration (mg/L)	Measuring Instrument
Color	colorless	-	Color	Brown yellowish	-
pH	7.7	HQ40d multi (Hach, Sydney, Australia)	pH	8.0	HQ40d multi(Hach, Sydney, Australia)
Turbidity, NTU	1.1	2100P Turbidimeter (Hach, Sydney, Australia)	Turbidity, NTU	35.0	2100P Turbidimeter Hach, Sydney, Australia)
Conductivity mS/cm	50.3	HQ14d Conductivity Hach, Sydney, Australia)	Conductivity ms/cm	12.1	HQ14d Conductivity Hach, Sydney, Australia)
Total dissolved solids (TDS)	32,800	-	TDS	4500	-
K^+^	505.8	7900 ICP-MS (Agilent, Auburn, United States)	Total organic carbon (TOC)	145.1 ± 5	TOC analyzer (Shimadzu CorporationTokyo, Japan)
Cl^−^	7177.4	7900 ICP-MS (Agilent, Auburn, United States)	TSS	27–117	(Agilent, Auburn, United States)
Na^+^	11,952.2	7900 ICP-MS (Agilent, Auburn, United States)	Total irons	3.5–52	(Agilent, Auburn, United States)
Ca^2+^	624.3	7900 ICP-(Agilent, Auburn, United States)	Ammonia	<0.5	5051—Ammonium Flow Plus ISE
SO_4_^2−^	2315.3	DIONEX AS-AP (ThermoFisher Sydney, Australia)	Ca^2+^	126 ± 5	(Agilent, Auburn, United States)
Mg^2+^	1383.6	(Agilent, Auburn, United States)	Mg^2+^	95.3 ± 5	(Agilent, Auburn, United States)
-	-	-	K^+^	47.87	(Agilent, Auburn, United States)

**Table 2 membranes-12-00951-t002:** Membrane specifications as provided by the manufacturer.

Characteristics	Values
Nominal pore size, µm	0.45
Thickness, µm	184 ± 38.5
Bubble point, psi	12.3 ± 0.725
Contact angle	129° ± 2°

**Table 3 membranes-12-00951-t003:** Elemental composition (wt%) of pristine and fouled membranes by seawater and landfill leachate.

Element	Pristine Membrane	Seawater-Fouled	Landfill Leachate after 15 h of Fouling	Landfill Leachate after 30 h of Fouling
C	93.7	-	70.42	44.01
Na	2.81	6.91	1.86	2.45
Mg	1.73	5.82	0.90	1.42
Cl	0.60	0.39	0.19	0.29
K	0.33	0.72	0.20	0.29
Ca	0.26	3.6	0.20	0.39
Fe	0.57	-	0.66	2.06
O	-	73.54	-	-
S	-	12.29	-	-
N	-	-	24.27	50.39

**Table 4 membranes-12-00951-t004:** Pore size and contact angle analysis of pristine and fouled membranes.

Membrane Type	Smallest Pore Diameter (μm)	Largest Pore Diameter (μm)	Mean Pore Diameter (μm)	Contact Angle (°)
Pristine membrane	0.213 ± 0.010	0.296 ± 0.009	0.248 ± 0.008	129 ± 2
Seawater-fouled 65 h	0.201 ± 0.008	0.294 ± 0.008	0.231 ± 0.008	115 ± 3
Landfill leachate-fouled 15 h	0.182 ± 0.008	0.296 ± 0.007	0.194 ± 0.007	102 ± 3
Landfill leachate-fouled 30 h	0.166 ± 0.006	0.244 ± 0.007	0.170 ± 0.008	93 ± 3

## Data Availability

The data presented in this study are available on request from the corresponding author.

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
