# Peer review of "Fouling and Performance Investigation of Membrane Distillation at Elevated Recoveries for Seawater Desalination and Wastewater Reclamation"

_membranes, 2022, doi:10.3390/membranes12100951_

Round 1

Reviewer 1 Report

Summary:

In this work, the authors studied the seawater and landfilled leachate treatments using the air gap membrane distillation (AGMD) process. The focus is on the impact of elevated recovery rates (i.e., 80%, 85%, and 90%) on the fouling and performance of the commercial PTFE membrane, which is applied to the MD process. The highest water permeance decline was observed with a 90% water recovery at 65 °C feed temperature. The declines in the water permeance and rejection are ascribed to membrane fouling, which is examined and analyzed by using various techniques such as SEM, SEM/EDX and FTIR. In addition, the authors also developed membrane cleaning methods using ID water and H2O2 solution at different temperatures. Generally, it is a good case study of AGMD process using real wastewaters. However, the writing and presentation of this manuscript need to be improved significantly. It is suitable for publishing in Membranes after addressing the following issues and improving the manuscript. 

List of comments:

1.      The English command and writing need to be improved. Please do careful and thorough proof reading.

2.      In the Abstract, the "PTFE" acronym should be placed before the "polymer"

3.      Table1, it would be better to have a solid line or alike to separate the seawater and landfill leachate blocks. Otherwise, the table contents look messy and confusing.

4.      Figure1, in the second image (from top to bottom, PVDF membrane was indicated. Is the PVDF membrane also used in this work? If not, please remove PVDF.

5.      Page 4, the definition of fouling factor is a bit confusing. Ja is the DI water flux after fouling or after cleaning. It seems that the former is more reasonable. Please verify and clarify.

6.      Equation 3, the rejection usually is expressed in percentage, but in Equation 3, the % sign is missing.

7.      The recovery rate means the total pure water extracted from the feed, then divided by the raw feed. The definition of recovery rate seems wrong in the equation 4. Equation 4 is literally and mathematically the same with Equation 3. Please verify and clarify.

8.      Please define and give the full names of FESEM, EDX and all other acronyms at the places where they are first used.

9.      Page 7, statement: “…were investigated for membrane cleaning at the end of each DCMD…”.  The statement is confusing. Is it the end of DCMD or AGMD? You did AGMD study, but said DCMD. Please verify and clarify.  

10.  Please improve the figures’ quality and presentation. For example, Figure 2 B and D not aligned well in vertical direction. The captions in the sub-figures are small and not easy to read.

11.  Figure 4, caption should be "Fouling factor and rejection".

12.  Page 9, where is the Table 4? Please place and present the Table at or near the place where you first mention it. Otherwise, readers have to go somewhere to look for it, causing inconvenience.

13.  The presentation of Figure 8 is poor and not professional. There is no 'E' label in the FTIR figure. It is suggested that SEM images can be grouped as a figure and FTIR can be a separated figure.

14.  Page 13, “were 0.213 ± 0.010 xm”. The unit is totally unreadable.

15.  Table 4, the unit is missing in this 'Largest pore diameter' column.

16.  Page 14, statement: “…EDX results indicate a potential for combined organic and inorganic 407 fouling due to the detection of C, Mg, and Ca on the membrane surface (Table 3).”. It is not "due to", it should be 'evidenced by' or likewise.

17.  Figure 9, small letters are used inside the figure while capital letters are used in the caption. Please make the presentation consistent.

18.  This work studies the fouling issues of MD. However, in the Introduction section, the authors rarely mention fouling issues and how to overcome the fouling issues conducted/investigated by the others in the past. Please clarify and improve the Introduction.

19.  Traditionally, as described in the Introduction section, the water recovery rate of seawater RO is about 50%. However, with the new process of osmotically assisted reverse osmosis (OARO), the water recovery rate of seawater RO process could be up to 80%. The following paper cover the topics of membranes development for seawater desalination via RO and OARO: Ultra-strong polymeric hollow fiber membranes for saline dewatering and desalination, Nature Communications, 12 (2021) 2338. Recent membrane development for high-performance MD applications and MD fouling control can be found in the following paper: Mechanically Strong Tri-bore Hollow Fiber Membranes with Janus Pores for Anti-wetting and Anti-fouling Membrane Distillation, Nature Communications, Chemical Engineering Journal, 429 (2022) 132455. These 2 papers may be informative and helpful to improve this work, especially for the Introduction section. You may include and cite these papers to improve your study and manuscript.

20.  Please indicate novelty of this study, and the contribution of this study to the field of water treatment by membrane technologies. Such novelty and contribution statements should, and especially needed in the Conclusions and/or Introduction sections.

Author Response

Summary:

In this work, the authors studied the seawater and landfilled leachate treatments using the air gap membrane distillation (AGMD) process. The focus is on the impact of elevated recovery rates (i.e., 80%, 85%, and 90%) on the fouling and performance of the commercial PTFE membrane, which is applied to the MD process. The highest water permeance decline was observed with a 90% water recovery at 65 °C feed temperature. The declines in the water permeance and rejection are ascribed to membrane fouling, which is examined and analyzed by using various techniques such as SEM, SEM/EDX and FTIR. In addition, the authors also developed membrane cleaning methods using ID water and H2Osolution at different temperatures. Generally, it is a good case study of AGMD process using real wastewaters. However, the writing and presentation of this manuscript need to be improved significantly. It is suitable for publishing in Membranes after addressing the following issues and improving the manuscript. 

Answer: The authors would like to thank Reviewer 2 for the comments to improve the manuscript. All the changes are highlighted in red colour in the manuscript with the page number given in the responses below.

List of comments:

Editor and Reviewer comments:

  1. Reviewer 1: The English command and writing need to be improved. Please do careful and thorough

Answer: Thanks; the English language has been checked in the revised manuscript

Action: multiple changes have been included in the edited manuscript to convey the meanings. All the changes are highlighted in red colour in the manuscript.

  1. Reviewer 1: In the Abstract, the "PTFE" acronym should be placed before the "polymer".

Answer: Thank you, “PTFE” was placed before polymer in the abstract.

Action: Please see Page 1, line 20.

This study reports on the impact of elevated recovery rates (i.e., 80%, 85%, and 90%) on the fouling and performance of Air gap membrane distillation (AGMD) with real seawater and landfill leachate wastewater using polytetrafluoroethylene (PTFE) polymer membrane.

  1. Reviewer 1: Table1, it would be better to have a solid line or alike to separate the seawater and landfill leachate blocks. Otherwise, the table contents look messy and confusing.

Answer: Thank you, Table (1) has been redesigned to separate seawater information from the landfill leachate wastewater.

Action: Please see Page 3, Line 113.

Additionally, all tables were redesigned in the manuscript for consistency.

  1. Reviewer 1: Figure1, in the second image (from top to bottom, PVDF membrane was indicated. Is the PVDF membrane also used in this work? If not, please remove PVDF.

Answer: Thank you, in figure 1, PVDF was removed, and the image was edited.

Action: Please see Page 5, Line 160.

  1. Reviewer 1: Page 4, the definition of fouling factor is a bit confusing. Jais the DI water flux after fouling or after cleaning. It seems that the former is more reasonable. Please verify and clarify.

Answer: Thank you, Ja is the DI water flux after fouling, as indicated in the manuscript.

Action: Please see Page 4, Line 147.

 Ja is the DI water flux after experiment time (fouled MD) and before chemical cleaning.

  1. Reviewer 1: Equation 3, the rejection usually is expressed in percentage, but in Equation 3, the % sign is missing.

Answer: Thank you, the percentage sign was added  

Action: Please see Page 4, Line 156.

The rejection (R %) was calculated using Eq. (3):

  1. Reviewer 1: The recovery rate means the total pure water extracted from the feed, then divided by the raw feed. The definition of recovery rate seems wrong in the equation 4. Equation 4 is literally and mathematically the same with Equation 3. Please verify and clarify.

Answer: Thank you, equation 4 was corrected

Action: Please see Page 5, Lines 164-165.

The recovery rate (Rr %) is estimated as the ratio of permeate flow to the feed flow according to the following expression:

(4)

Where, Qp and Qf are the permeate and feed solutions flow rates (L/h), respectively.

  1. Reviewer 1: Please define and give the full names of FESEM, EDX and all other acronyms at the places where they are first used.

Answer: Thank you, these have been defined in the manuscript prior to the method section.

Action: Please see Page 3, Line 98-99

  1. Reviewer 1: Page 7, statement: “…were investigated for membrane cleaning at the end of each DCMD…”.  The statement is confusing. Is it the end of DCMD or AGMD? You did AGMD study, but said DCMD. Please verify and clarify.  

Answer: Thank you, the paragraph was edited, and DCMD was replaced with AGMD.

Action: Page 8, Line 254.

The DI water at 55 oC or 65 oC (depending on the feed temperature of the experiment) and H2O2 solution were investigated for membrane cleaning at the end of each AGMD experiment.

  1. Reviewer 1: Please improve the figures’ quality and presentation. For example, Figure 2 B and D not aligned well in vertical direction. The captions in the sub-figures are small and not easy to read.

Answer: Thank you, Figure 2 has been updated.

Action: Please see Page 7, Line 225.

  1. Reviewer 1: Figure 4, caption should be "Fouling factor andrejection".

Answer: Thank you, the caption has been updated accordingly.

Action: Please see Page 9, line 284.

  1. Reviewer 1: Page 9, where is the Table 4? Please place and present the Table at or near the place where you first mention it. Otherwise, readers have to go somewhere to look for it, causing inconvenience.

Answer: All the tables have been updated and moved close to the places where they were first mentioned in the manuscript.

Action: Please see Page 10, Line 316.

  1. Reviewer 1: The presentation of Figure 8 is poor and not professional. There is no 'E' label in the FTIR figure. It is suggested that SEM images can be grouped as a figure and FTIR can be a separated figure.

Answer: Thank you, the FT-IR has been removed to a separate figure, Figure 9.

Action: Please see Page 14, Line 391..

  1. Reviewer 1: Page 13, “were 0.213 ± 0.010 xm”. The unit is totally unreadable.

Answer: Thank you, all units in section 3.5.2 have been corrected.

Action: Please see Page 14, Line 397-405.

  1. Reviewer 1: Table 4, the unit is missing in this 'Largest pore diameter' column.

Answer: Thank you, the units were added to the Largest pore diameter and Mean pore diameter (μm)

Action: Please see Page 11, Line 317

  1. Reviewer 1: Page 14, statement: “…EDX results indicate a potential for combined organic and inorganic 407 fouling due to the detection of C, Mg, and Ca on the membrane surface (Table 3).”. It is not "due to", it should be 'evidenced by' or likewise.

Answer: Thank you, the paragraph was edited.

Action: Please see Page 14, Line 430-432

For the landfill leachate experiments, EDX results indicate a potential for combined organic and inorganic fouling, evidenced by the detection of C, Mg, and Ca on the membrane surface (Table 3).

  1. Reviewer 1: Figure 9, small letters are used inside the figure while capital letters are used in the caption. Please make the presentation consistent.

Answer: Thank you, Figure 9, now Figure 10 has been updated.

Action: Please see Page 16, Line 465.

  1. Reviewer 1: This work studies the fouling issues of MD. However, in the Introduction section, the authors rarely mention fouling issues and how to overcome the fouling issues conducted/investigated by the others in the past. Please clarify and improve the Introduction.

Answer: Thank you, the introduction has been improved by incorporating new studies from the literature.

Action: Please see Page 2, Line 44, Line 59-62, 67-68, 85-93.

  1. Reviewer 1: Traditionally, as described in the Introduction section, the water recovery rate of seawater RO is about 50%. However, with the new process of osmotically assisted reverse osmosis (OARO), the water recovery rate of the seawater RO process could be up to 80%. The following paper cover the topics of membranes development for seawater desalination via RO and OARO: Ultra-strong polymeric hollow fiber membranes for saline dewatering and desalination, Nature Communications, 12 (2021) 2338. Recent membrane development for high-performance MD applications and MD fouling control can be found in the following paper: Mechanically Strong Tri-bore Hollow Fiber Membranes with Janus Pores for Anti-wetting and Anti-fouling Membrane Distillation, Nature Communications, Chemical Engineering Journal, 429 (2022) 132455.These 2 papers may be informative and helpful to improve this work, especially for the Introduction section. You may include and cite these papers to improve your study and manuscript.

Answer: Thank you, these studies have been incorporated and referred to in the updated version.

Action:

: Please see Page 2, Line 44, Line 59-62, 67-68, 85-87.

  1. Reviewer 1: Please indicate novelty of this study, and the contribution of this study to the field of water treatment by membrane technologies. Such novelty and contribution statements should, and especially needed in the Conclusions and/or Introduction sections.

Answer: Thank you, the edited manuscript included more data and information in the introduction section to underline the work's novelty and research gap.

Action: Please see Page 2, Line 490-494

Introduction section: Despite the large research output in AGMD technology, no study investigated the MD membrane fouling at elevated recovery rates for water reclamation. Most studies investigated the AGMD fouling at normal recovery rates. High recovery rates are particularly important in desalination and wastewater treatment to reduce brine discharge and metal ions reclamation. Also, the fouling mechanisms of the AGMD system vary with the feed type, and there are no data to compare seawater and wastewater fouling mechanisms at elevated recovery rates.

Conclusion section: Future work should investigate the impact of membrane material on the performance of the AGMD for the treatment of seawater and wastewater at elevated recovery rates to identify the suitability of the membrane for the process. Also, a large field test will be desirable to demonstrate the AGMD performance when scaled up.

Reviewer 2 Report

In this manuscript, the impact of elevated recovery rates (i.e., 80%, 85%, and 90%) on

the fouling and performance of Air gap membrane distillation (AGMD) with real seawater and landfill leachate wastewater using polytetrafluoroethylene polymer (PTFE) membrane was reported. Some interesting conclusions were obtained. However, based on the following comments, I don’t think that this manuscript reaches the level for its publication in Membranes.

1.     In Introduction Section, the introduction in first paragraph of RO is redundant.

2.     The research value of MD performance at high recovery rates needs to be clarified

3.     How to explain the phenomenon of increase in average flux when the recovery rate increased to 90%?

4.     Why the average fluxes at the same temperature are different in the experiments conducted to 80%, 85% or 90% recovery.

5.     Lack of error bars in Figure 2, Figure 3, Figure 5 and Figure 6.

6.     How to explain the inverse correlation between permeate TDS and recovery rate?

7.     Lack of scale bar in Figure 8 (A)~(D)

8.     The font sizes in Figure 9 (c) and (d) are not uniform, and the histogram of Figure 9 (f) is not centered. Further, it is difficult to visually see the difference in the rejection rates of seawater and wastewater at 55°C and 65°C.

Author Response

In this manuscript, the impact of elevated recovery rates (i.e., 80%, 85%, and 90%) on

the fouling and performance of Air gap membrane distillation (AGMD) with real seawater and landfill leachate wastewater using polytetrafluoroethylene polymer (PTFE) membrane was reported. Some interesting conclusions were obtained. However, based on the following comments, I don’t think that this manuscript reaches the level for its publication in Membranes.

Authors response: The authors would like to thank the reviewer for their comments and suggestions for improving the quality of the work.

  1. In Introduction Section, the introduction in first paragraph of RO is redundant.

Response: The author introduced the RO in paragraph 1, to set the grounds for development in MD research. Since RO is the most widely used technology, one has to justify the needs for research in MD which is the only technology comparable to RO in terms of rejection.

Action: Please see Page 2, Line 59-62

  1. The research value of MD performance at high recovery rates needs to be clarified

Action: Please see Page 2, Line 85-93.

  1. How to explain the phenomenon of increase in average flux when the recovery rate increased to 90%?

Response: Since 90% recovery time was of shorter duration as mentioned, therefore, the average flux obtained was higher than the other recoveries.

Action: Please see Page 6, Line 201-205

  1. Why the average fluxes at the same temperature are different in the experiments conducted to 80%, 85% or 90% recovery.

Response: As the wastewater and seawater were real feed solutions, slightly different results are observed as the property of the feed solution are not consistent, for example, due to settling of the feed solution over time.

  1. Lack of error bars in Figure 2, Figure 3, Figure 5 and Figure 6.

Response: Error bars are only added to the bar charts, adding error bars to the line graphs will make it hard to read and interpretation would be harder.

  1. How to explain the inverse correlation between permeate TDS and recovery rate?

Action: Please see Page 244-246

  1. Lack of scale bar in Figure 8 (A)~(D)

Did the reviewer meant error bars? As it seems it is SEM pictures and FT_IR analysis.

  1. The font sizes in Figure 9 (c) and (d) are not uniform, and the histogram of Figure 9 (f) is not centered. Further, it is difficult to visually see the difference in the rejection rates of seawater and wastewater at 55°C and 65°C.

Figure 9 has been updated in the manuscript. Please see Page 16, Line 465.

Reviewer 3 Report

Some examples of the weaknesses of this article will be given below and are not intended to be an
exhaustive analysis:

1. Please revise the introduction to emphasize the work's original contribution, the scientific issue
that the study addresses and the research gap/problem found.

2. Highlight the novelty of the study in the last part of the introduction.

3. How has controlled the essay’s temperature and flow rate?

4. Since it is experimental research, it is generally necessary to show the experimental device
picture in the paper along with the schematic of the whole assembled set-up including at least
the test cell, pumps, heating, cooling, and temperature measuring elements.

5. Dimensions should be written in Fig. 1 and correct PVDE to PVDF

6. Make sure to use the same font and line spacing throughout the paper.

7. Recommend future scope of the study.

8. The following latest and related work are recommended to improve the study:

https://doi.org/10.1016/j.jenvman.2021.113922
, https://doi.org/10.3390/separations9010001,
https://doi.org/10.1016/j.memsci.2021.119916
, https://doi.org/10.1080/10643389.2021.1877032,
https://doi.org/10.3390/w13223241
, https://doi.org/10.1016/j.jwpe.2022.102615,
9.
All the references must be revised and cited in accordance with the instructions for authors.

Author Response

Some examples of the weaknesses of this article will be given below and are not intended to be an
exhaustive analysis:

The authors would like to thank Reviewer 2 for the comments to improve the manuscript. All the changes are highlighted in red color in the updated manuscript.

  1. Reviewer 2: Please revise the introduction to emphasize the work's original contribution, the scientific issue that the study addresses, and the research gap/problem found.

Answer: Thank you, Authors have addressed reviewer comments in the revised manuscript to underline the importance and novelty of the research.

Action: Please see Page 2, Line 44-45, 59-62,66-68,85-93

  1. Reviewer 2: Highlight the novelty of the study in the last part of the introduction.

Answer: Thank you, authors have included the new discussion in the introduction to highlight the importance of the research in the last paragraph.

Action: Please see Page 2, Line 88-93

The following paragraph has been added on page 2 “Despite the large research output in AGMD technology, no study investigated the MD membrane fouling at elevated recovery rates for water reclamation. Most studies investigated the AGMD fouling at normal recovery rates. High recovery rates are particularly important in desalination and wastewater treatment to reduce brine discharge and metal ions reclamation. Also, the fouling mechanisms of the AGMD system vary with the feed type, and there are no data to compare seawater and wastewater fouling mechanisms at elevated recovery rates.”

  1. Reviewer 2: How has controlled the essay’s temperature and flow rate?

Answer: The feed solution was placed on a hot plate with a thermostat to control the temperature, whereas the flow rate was measured using flow meter.

Action: Please see Page 4, Line 130-133.

  1. Reviewer 2 Since it is experimental research, it is generally necessary to show the experimental device picture in the paper along with the schematic of the whole assembled set-up including at least
    the test cell, pumps, heating, cooling, and temperature measuring elements.

Answer: Thank you, image was added to the Appendix.

Action: Please see Page 17, Line 517.

  1. Reviewer 2: Dimensions should be written in Fig. 1 and correct PVDE to PVDF

Answer: Thank you, the figure was updated and MD corrected to PTFE in the figure.

Action: Please see Page 5, Line 160.

  1. Reviewer 2: Make sure to use the same font and line spacing throughout the paper.

Answer: Thank you, authors checked the font size and line spacing to be identical throughout the manuscript.

Action: The authors have formatted the manuscript accordingly.

  1. Reviewer 2: Recommend the future scope of the study.

Answer: Thank you, the edited manuscript discussed future work in the field and potential scale up of the experimental work.

Action: Please see Page 17, Line 501-505

“Future work should investigate the impact of membrane material on the performance of the AGMD for the treatment of seawater and wastewater at elevated recovery rates to identify the suitability of the membrane for the process. Also, a large field test will be desirable to demonstrate the AGMD performance when scaled up.”

  1. Reviewer 2: The following latest and related work are recommended to improve the study:
    https://doi.org/10.1016/j.jenvman.2021.113922, https://doi.org/10.3390/separations9010001,
    https://doi.org/10.1016/j.memsci.2021.119916, https://doi.org/10.1080/10643389.2021.1877032,
    https://doi.org/10.3390/w13223241, https://doi.org/10.1016/j.jwpe.2022.102615,

Answer: Thank you, authors have added some of the recommended articles as requested by the reviewer to improve the work.

Action:  Please see Page 2, Line 44, 46, Line 59-62, 67-68, 85-87 and reference list.

  1. Reviewer 2: All the references must be revised and cited in accordance with the instructions for authors.

Answer: Thank you,

Action: The references are revised as per the journal guidelines.

Round 2

Reviewer 1 Report

The authors have addressed all the concerns and issues in details. The revised version is suitable for publishing. One minor comment: Figure A.1. Schematic diagram of the AGMD setup, the background of the figure is messy and caused confusion. It would be better if the background was removed and covered. 

Reviewer 3 Report

The authors have addressed most of my comments satisfactorily. The manuscript is acceptable for publication after the following 2 minor revisions.

1. Please indicate the peaks and their functioning groups in Fig. 9 and if possible, discuss them in the related text.

2. Please ensure the consistency of references according to Membranes guidelines.